# Impact on clinical outcomes of renin-angiotensin system inhibitors against doxorubicin-related toxicity in patients with breast cancer and hypertension: A nationwide cohort study in South Korea

**Hui-Jeong Hwang** [1], **Taek-Gu Lee** [2]*

1 Department of Cardiology, Kyung Hee University Hospital at Gangdong, Kyung Hee University College of Medicine, Seoul, Korea, 2 Department of Surgery, Chungbuk National University Hospital, Chungbuk National University College of Medicine, Cheongju, Korea

* ninehouse@hanmail.net

## Abstract

### Background

Although doxorubicin (DOX) is a commonly used potent chemotherapeutic agent in patients with breast cancer, its cardiotoxic effect is a concern, particularly in patients with hypertension. Antihypertensive renin-angiotensin system (RAS) inhibitors may potentially play a role in preventing overt heart failure (HF) due to DOX toxicity. This study aimed to evaluate whether the use of RAS inhibitors improves clinical outcomes in patients with hypertension and breast cancer undergoing DOX-containing chemotherapy.

### Methods

A total of 54,344 female patients who were first diagnosed with breast cancer and initiated into DOX therapy between 2008 and 2015 were recruited from a nationwide Korean cohort. Patients were divided into two groups: with and without hypertension (HT, n = 10,789; non-HT, n = 43,555), and the RAS inhibitor group (n = 1,728) was sub-classified from the HT group. Two propensity score-matched cohorts were constructed to compare the clinical outcomes between non-HT and HT groups and between non-HT and RAS inhibitor groups. The primary outcome was the composite of HF and death.

### Results

After propensity score matching, the HT group had a higher risk for HF (adjusted hazard ratio [HR] = 1.30, 95% confidence intervals [95% CI] = 1.09–1.55) compared to the non-HT group, but there was no significant difference in primary outcome between the two groups. The RAS inhibitor group had a lower risk for primary outcome (adjusted HR = 0.78, 95% CI = 0.65–0.94) and death (adjusted HR = 0.81, 95% CI = 0.66–0.99) compared to the non-HT group.

**Data Availability Statement:** Data supporting this study are publicly available on the HIRA website (https://opendata.hira.or.kr). In addition, all relevant

data are within the manuscript and its Supporting Information files.

**Funding:** HJH No. 2022-01-01 the Working Group on Cardio-Oncology of the Korean Society of Cardiology https://www.cardiooncology.or.kr/ The funders had no role in study design, data collection and analysis, decision to publish, or preparation of the manuscript.

**Competing interests:** The authors have declared that no competing interests exist.

## Conclusions

Hypertension is a risk factor for HF in patients with breast cancer undergoing DOX chemo-therapy. However, the RAS inhibitors used to treat hypertension may contribute to decreased mortality and improved clinical outcomes.

## Introduction

Doxorubicin (DOX) is a potent chemotherapeutic agent commonly used for patients with breast cancer. However, they are dose-dependently cardiotoxic and can cause heart failure (HF) even at a low-dose in patients with concomitant use of the human epidermal growth factor receptor-2 inhibitor trastuzumab or with high cardiovascular risk factors [1, 2]. Hypertension is a risk factor for DOX-related cardiotoxicity [1, 3, 4]; however, patients with well-controlled hypertension may have better clinical outcomes than those without hypertension [5]. Therefore, treatment of hypertension is especially important in patients with cancer.

The prophylactic use of renin-angiotensin system (RAS) inhibitors (renin-angiotensin-converting enzyme inhibitors and angiotensin receptor blockers) before DOX treatment decreased DOX-induced apoptosis and improved survival rates in animal models [6, 7]. Moreover, RAS inhibitors reduced DOX-related cardiac dysfunction assessed by left ventricular ejection fraction (LVEF) in clinical studies [8, 9]. However, recent meta-analyses have demonstrated that the use of RAS inhibitors does not significantly reduce the risk of overt HF and mortality and increases the risk of hypotension in patients with cancer receiving anthracycline-containing chemotherapies [10, 11]. Furthermore, RAS inhibitors are recommended as first-line antihypertensive drugs in the guidelines [1], but improved outcomes have been demonstrated in some solid tumors, including those of the urinary tract, colon, pancreas, and prostate, but not in the breast [12–17]. In our recent study, RAS inhibitors had better cardiovascular outcomes than beta blockers or thiazide and thiazide-like diuretics in patients with hypertension and breast cancer undergoing DOX therapy, but similar outcomes to calcium channel blockers [18]. In this study, we evaluated whether the use of RAS inhibitors as antihypertensive agents improves clinical outcomes in patients with breast cancer undergoing DOX-containing chemotherapy compared to those without hypertension, using Korean nationwide cohort data.

## Methods

### Data sources

This study used nationwide claims data from the Health Insurance Review and Assessment (HIRA) Service database of South Korea, which provides demographic and diagnostic data based on the International Classification of Disease-10th Revision-Clinical Modification diagnostic codes, electronic data interchanging overall medical service information, including medications, procedures, and operations, and death information of the entire Korean population covered by medical care (approximately 45 million).

### Study population

As described in our previous study [18], female patients aged 17–70 years who were first diagnosed with breast cancer and started DOX therapy between January 1, 2008 and December 31, 2015 were recruited (Fig 1). The index date was defined as the date on which DOX treatment was initiated.

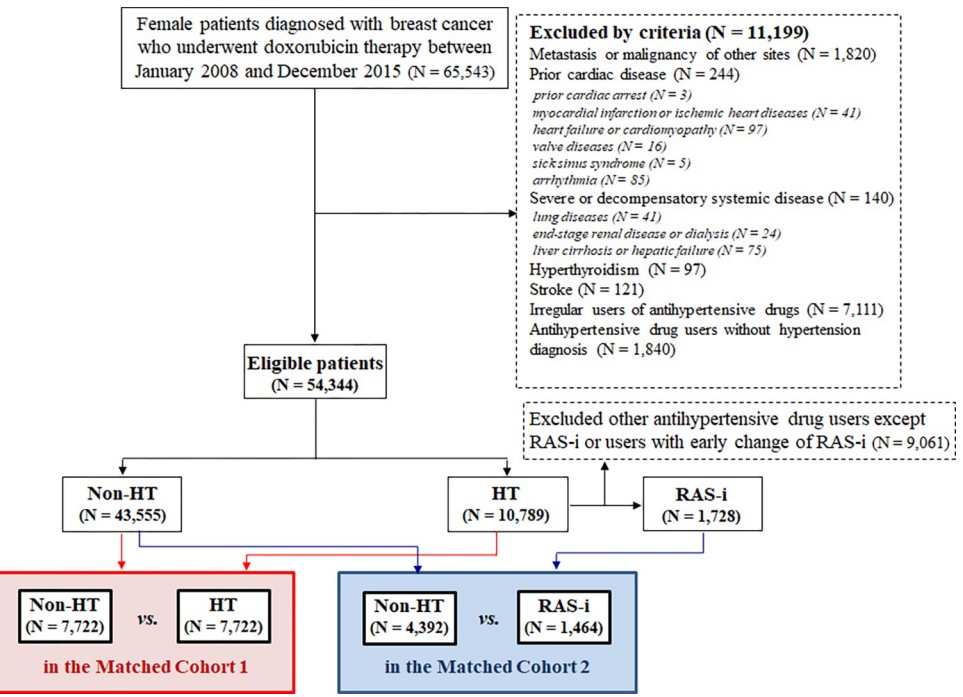

**Fig 1. Study flow chart.** Non-HT, without hypertension; HT, hypertension; RAS-i, renin-angiotensin system inhibitor.

The following patients were excluded: 1) patients diagnosed with breast cancer before 2008; 2) patients who had metastasis or malignancy of other sites on the index date; 3) patients with significant heart diseases including prior cardiac arrest, myocardial infarction or ischemic heart diseases, HF or cardiomyopathy, valvular heart diseases, sick sinus syndrome, and arrhythmias such as atrial/ventricular flutter or fibrillation; 4) patients with severe systemic disease, including severe lung diseases, end-stage renal disease or dialysis, and liver cirrhosis or hepatic failure, or prior history of decompensatory condition; 5) patients with hyperthyroidism or stroke; and 6) patients with inaccurate information to diagnose hypertension.

Patients were followed up until the end date of December 31, 2020. This study was approved by the ethics committee of our hospital (KHNMC 2021-09-020). The requirement for informed consent was waived because the provided claims data were anonymized and de-identified. Data supporting this study are publicly available on the HIRA website (https://opendata.hira.or.kr).

### Study design, covariates, and clinical outcomes

The patients were divided into two groups: those with and without hypertension (HT and non-HT groups, respectively). In the HT group, patients who received RAS inhibitors as monotherapy for at least three months from the index date were sub-classified into the RAS inhibitor group. Patients who received RAS inhibitors after the index date were excluded. The diagnostic definitions of comorbidities, including hypertension, diabetes mellitus, dyslipidemia, and angina, are described in S1 Table. The total amount of DOX used in each patient was calculated and high-dose DOX was defined as > 4 cycles of DOX or a cumulative dose of > 240 mg/m$^2$ [1].

Two propensity score-matched cohorts were used to compare the clinical outcomes as follows: 1) non-HT vs. HT groups in cohort 1 and 2) non-HT vs. RAS inhibitor groups in cohort

2. The primary outcome was the composite of HF and death. The secondary outcomes were HF and death. Detailed definitions of the outcomes are provided in S1 Table.

## Statistical analysis

The baseline clinical characteristics of the HT and RAS inhibitor groups were compared with those of the non-HT group in the overall study population. Continuous variables are expressed as mean ± standard deviation and were compared using Student's t-test; categorical variables are expressed as group percentages and were compared using the chi-square test. The crude event numbers and incidence rates of clinical outcomes were calculated for each group. The incidence is expressed per 100 person-years of follow-up. Before comparing the clinical outcomes, propensity score matching was performed to balance the baseline clinical characteristics between the two groups using the greedy nearest-neighbor algorithm. The covariates included age, indexed year, history of diabetes mellitus, dyslipidemia, angina, and use of antithrombotic agents and statins. The matching rates were determined based on the sample size between the groups as follows: a 1:1 ratio between the non-HT and HT groups and a 3:1 ratio between the non-HT and RAS inhibitor groups. A standardized mean difference ≤ 0.1 for a covariate was defined as well-balanced. In each propensity score-matched cohort, the cumulative incidence rates for the primary outcome and HF were compared between the two groups using the log-rank test and described using Kaplan–Meier plots. Clinical outcomes were compared using Cox proportional hazards models and presented as crude and adjusted hazard ratios (HRs) and 95% confidence intervals (95% CIs). The covariates used for adjustment were consistent with those used during propensity score matching. For the sensitivity test, the 1-year composite outcomes were compared for each propensity score-matched cohort. Statistical analyses were performed using the R software version 3.5.1. Propensity score-matched models were constructed using the SAS Enterprise Guide version 7.15 (SAS Institute Inc., Cary, NC, USA). Statistical significance was set at $p < 0.05$.

## Results

### Baseline clinical characteristics and event rates for clinical outcomes in the overall population

Altogether 54,344 patients were enrolled (Fig 1). Among them, the number of patients without and with hypertension were 43,555 and 10,789, respectively. Among those with hypertension, 1,728 patients received RAS inhibitors. Patients with hypertension were older and had a higher incidence of diabetes mellitus, dyslipidemia, and angina compared to those without hypertension along with a history of taking antithrombotic agents and statins (Table 1). The use of high-dose DOX was more in patients without hypertension, while concomitant trastuzumab use was higher in patients with hypertension. Similarly, the RAS inhibitor group was older and had a higher incidence of comorbidities, increased use of antithrombotic agents, statins, and concomitant trastuzumab and radiotherapy in the left chest field, and reduced use of high-dose DOX than the non-HT group.

The incidence rates of primary outcomes were 1.34, 1.83, and 1.45 per 100 person-years in the non-HT, HT, and RAS inhibitor groups, respectively (Table 2).

### Differential clinical outcomes in each propensity score-matched cohort

In propensity score-matched cohort 1, the non-HT and HT groups included 7,722 and 7,722 patients, respectively (Fig 1). In cohort 2, the non-HT and RAS inhibitor groups included 4,392 and 1,464 patients, respectively. The covariates were well-balanced in each matched

**Table 1. Baseline clinical characteristics in the overall study population.**

| Characteristics | Non-HT | HT | | p value for non-HT | |
|---|---|---|---|---|---|
| | | **Total** | **RAS-i** | **Total** | **RAS-i** |
| Subjects, n | 43,555 | 10,789 | 1,728 | | |
| Age, years | 47 ± 8 | 56 ± 8 | 55 ± 8 | < 0.001 | < 0.001 |
| Indexed year, n (%) | | | | 0.168 | < 0.001 |
| *2008–2010* | 12,830 (30) | 3,095 (29) | 347 (20) | | |
| *2011–2013* | 17,565 (40) | 4,348 (40) | 685 (40) | | |
| *2014–2015* | 13,160 (30) | 3,346 (31) | 696 (40) | | |
| Medical history, n (%) | | | | | |
| *Diabetes mellitus* | 1,108 (2.5) | 1,979 (18) | 401 (23) | < 0.001 | < 0.001 |
| *Dyslipidemia* | 2,757 (6) | 4,157 (39) | 780 (45) | < 0.001 | < 0.001 |
| *Angina* | 482 (1) | 939 (9) | 134 (8) | < 0.001 | < 0.001 |
| Medications, n (%) | | | | | |
| *Antithrombotic agents* | 1,052 (2) | 3,448 (32) | 492 (29) | < 0.001 | < 0.001 |
| *Statins* | 2,633 (6) | 3,958 (37) | 751 (44) | < 0.001 | < 0.001 |
| Chemotherapy, n (%) | | | | | |
| *High-dose DOX* | 8,911 (21) | 2,000 (19) | 279 (16) | < 0.001 | < 0.001 |
| *Concomitant TRA* | 9,115 (21) | 2,490 (23) | 450 (26) | < 0.001 | < 0.001 |
| RT in the Lt. chest field, n (%) | 1,534 (4) | 421 (4) | 78 (5) | 0.062 | 0.034 |

Non-HT, without hypertension; HT, hypertension; RAS-i, renin-angiotensin system inhibitor; Dox, doxorubicin; TRA, trastuzumab; RT, radiotherapy; Lt., left

model (S2 Table). After propensity score matching, there was no significant difference between the two groups in mean follow-up period and anticancer therapies, including high-dose DOX, concomitant trastuzumab use, and radiotherapy in the left chest field (S3 Table).

In the matched cohorts, the HT group had a higher cumulative incidence rate of HF than the non-HT group, but did not differ significantly in the primary outcome (Fig 2). The RAS inhibitor group had a lower cumulative incidence of primary outcome than the non-HT

**Table 2. Event and event rate in the overall study population and the comparative risks for clinical outcomes in propensity score-matched cohorts.**

| Clinical outcomes | Overall population | | Propensity score-matched cohorts | | | |
|---|---|---|---|---|---|---|
| | **Events/No** | **IR** | **Crude HR (95% CI)** | **p value** | **Adjusted HR (95% CI)** | **p value** |
| **Primary outcome = HF + Death** | | | | | | |
| Non-HT | 4,526/43,555 | 1.34 | 1 (1.00–1.00) | NA | 1 (1.00–1.00) | NA |
| HT | 1,503/10,789 | 1.83 | 1.03 (0.94–1.13) | 0.534 | 1.03 (0.95–1.13) | 0.534 |
| RAS-i | 183/1,728 | 1.45 | 0.79 (0.65–0.94) | 0.010 | 0.78 (0.65–0.94) | 0.009 |
| **HF** | | | | | | |
| Non-HT | 820/43,555 | 0.24 | 1 (1.00–1.00) | NA | 1 (1.00–1.00) | NA |
| HT | 499/10,789 | 0.61 | 1.29 (1.09–1.54) | 0.004 | 1.30 (1.09–1.55) | 0.004 |
| RAS-i | 46/1,728 | 0.36 | 0.75 (0.51–1.12) | 0.160 | 0.74 (0.50–1.10) | 0.142 |
| **Death** | | | | | | |
| Non-HT | 3,925/43,555 | 1.16 | 1 (1.00–1.00) | NA | 1 (1.00–1.00) | NA |
| HT | 1,146/10,789 | 1.37 | 0.99 (0.89–1.09) | 0.795 | 0.99 (0.90–1.10) | 0.904 |
| RAS-i | 149/1,728 | 1.17 | 0.80 (0.66–0.99) | 0.036 | 0.81 (0.66–0.99) | 0.037 |

No, numbers; IR, incidence rate per 100 person-years; HR, hazard ratio; CI, confidence interval; Non-HT, without hypertension; HT, hypertension; RAS-i, renin-angiotensin system inhibitor; NA, not applicable

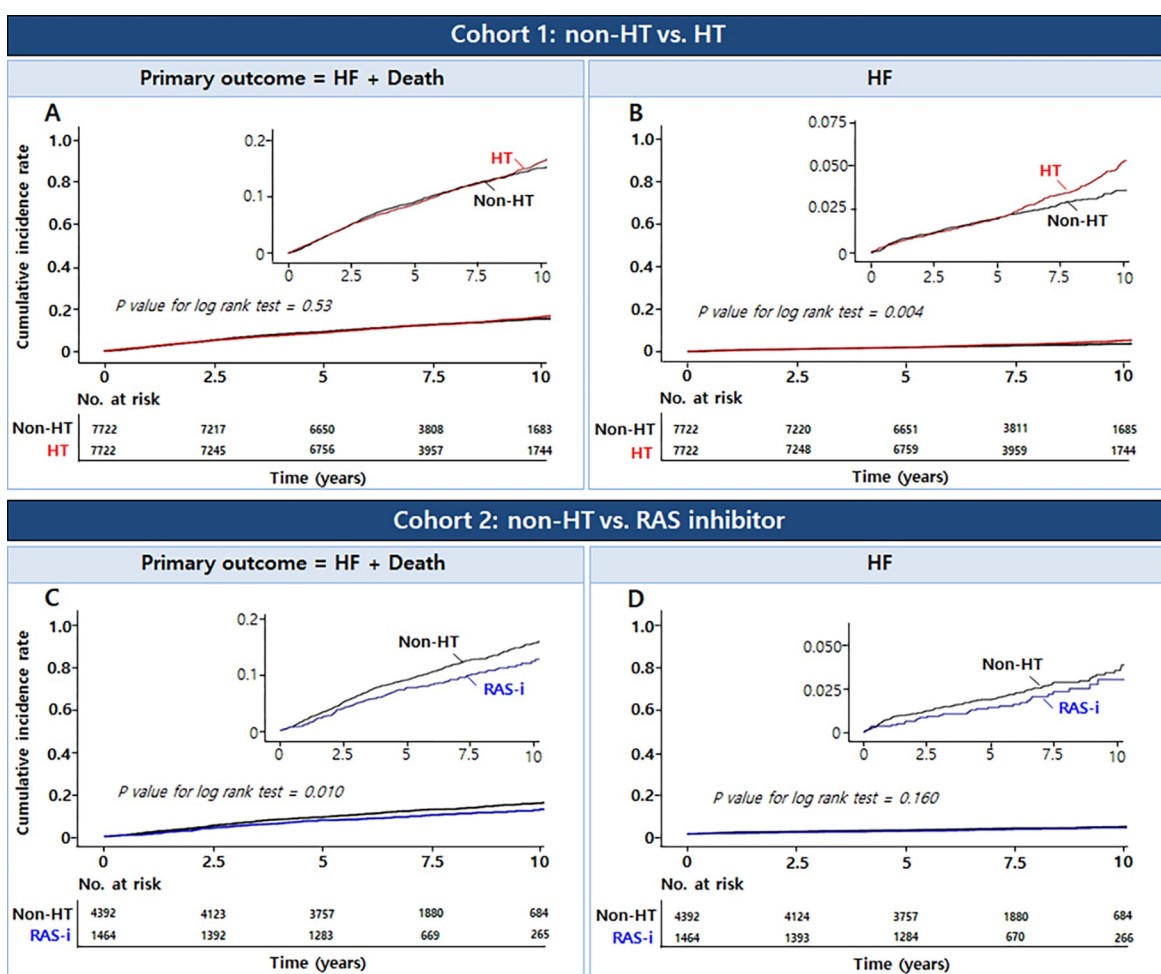

**Fig 2. Cumulative incidence curves for primary outcome and heart failure in each propensity score matched cohort.** HF, heart failure; No, number; Non-HT, without hypertension; HT, hypertension; RAS-i, renin-angiotensin system inhibitor.

group, but there was no significant difference in HF. After adjustment by covariates, the HT group had a higher risk for HF (adjusted HR [95% CI] = 1.30 [1.09–1.55], p = 0.004) compared to the non-HT group, but there was no significant difference in primary outcome between the two groups. The RAS inhibitor group had a lower risk for primary outcome (adjusted HR [95% CI] = 0.78 [0.65–0.94], p = 0.009) and death (adjusted HR [95% CI] = 0.81 [0.66–0.99], p = 0.037) than the non-HT group.

As a sensitivity test, the 1-year primary outcome was lower in the RAS inhibitor group (adjusted HR [95% CI] = 0.56 [0.33–0.95], p = 0.033) than that in the non-HT group (S4 Table). There was no significant difference in the 1-year primary outcome between the non-HT and HT groups.

## Discussion

Hypertension is a concurrent risk factor for HF [19, 20] and DOX-related cardiotoxicity [1, 3, 4]. However, there is no evidence that hypertension increases mortality in patients with cancer [21]. In this study, patients with hypertension had a higher risk of HF than those without hypertension, but the overall mortality was similar in both the groups. Consequently, the

primary outcome was not significantly different. In contrast, patients taking RAS inhibitors to treat hypertension had a risk of HF similar to those without hypertension, but had a lower mortality rate and consequently a lower primary outcome.

## Hypertension and DOX-related cardiotoxicity

Hypertension is a risk factor for DOX-related cardiotoxicity [1, 3, 4], and uncontrolled hypertension causes increased DOX-related HF [1, 5]. Szmit et al. demonstrated that hypertension was associated with more frequent LVEF decrease (19.7% in the HT group vs. 6.6% in the non-HT group, p = 0.004) in patients with non-Hodgkin's lymphoma undergoing DOX treatment [4]. Hershman et al. showed that hypertension increased the risk of DOX-related HF in an epidemiological study of patients with diffuse large B-cell lymphoma [3]. Similarly, in our study, hypertension was a risk factor for HF in patients with breast cancer undergoing DOX therapy.

## Cardioprotective effect of RAS inhibitors against DOX toxicity

In experimental models, DOX increased the activity of cardiac angiotensin-converting enzyme [22, 23] and angiotensin II (Ang II) [24, 25] and induced the overexpression of angiotensin II type 1 receptor (AT1R) [26, 27], causing oxidative stress, inflammation, fibrosis, and apoptosis of cardiomyocytes, resulting in cardiotoxicity [28]. Therefore, RAS inhibitors have been proposed as potential protective agents against DOX-induced cardiotoxicity. Indeed, the preemptive use of RAS inhibitors before DOX administration reduced the decrease in LVEF in several clinical studies [8, 9, 29]. However, recent meta-analyses found that RAS inhibitors did not significantly reduce the incidence of overt HF but rather increased the incidence of hypotension [10, 11]. Experts suggest this is because most trials mainly included patients with a low risk of HF; thus, larger randomized controlled trials are required for high-risk patients [1]. Unfortunately, we could not confirm an association between RAS inhibitor use and a reduced risk of HF.

## Impact on clinical outcomes of RAS inhibitors in patients undergoing anticancer therapies

Observational studies have demonstrated that RAS inhibitors improved overall mortality in patients with renal cell carcinoma undergoing vascular endothelial growth factor-targeting tyrosine kinase inhibitor (TKI) chemotherapy, particularly in those with TKI-induced hypertension [12–14]. TKIs cause hypertension in a dose-dependent manner as an on-target effect [30–32], thus TKI-induced hypertension is considered a predictor of better outcomes [33–36]. Furthermore, RAS inhibitors used in patients with TKI-induced hypertension have been proposed to reduce mortality due to their synergistic antiangiogenic effects with TKIs [13, 14].

In breast cancer tissues, the hyperactivated Ang II/AT1R axis plays an important role in promoting tumor growth, supporting tumor invasion and angiogenesis, and stimulating lymph node metastasis and cell migration [37–39]. Previous experimental studies have shown that RAS inhibitors inhibit breast cancer progression [39–41]. However, there is no clinical evidence of a survival benefit from the use of RAS inhibitors in patients with breast cancer [11, 15, 16]. In our study, RAS inhibitors used to treat hypertension during DOX therapy led to better clinical outcomes, including survival benefit. This positive finding may be attributed to the larger sample size and consequently, more events than those in previous studies [11, 15, 16].

## Limitations

This study had some limitations. First, the HF events in our study were fewer than the approximately 5% reported in previous studies [42, 43]. Because this study was based on claims data,

HF was defined as clinically symptomatic assessed by diagnostic codes and diuretic use, but not as asymptomatic HF, assessed by the decrease of LVEF. In addition, we excluded patients at risk of clinically misdiagnosed HF, including those with severe lung disease, liver failure or cirrhosis, and hyperthyroidism, and those with frail conditions, including age >70 years, who are at high risk for HF but likely have uncontrolled confounding factors. This may have resulted in fewer HF events and insignificant differences in HF risk between the RAS inhibitor and non-HT groups. Second, the HIRA Service database of South Korea does not provide clinical data, including cancer stage and blood pressure. According to statistical reports from the Korean Breast Cancer Society (https://www.kbcs.or.kr), the 5-year survival rates for Korean patients with breast cancer between 2001 and 2012 dropped rapidly to below 35% in stage IV, compared with those in stages I-III (above 90% in stages I-II, 75% in stage III). Therefore, we excluded patients with metastasis to minimize mortality unrelated to interventions, including chemotherapy and RAS inhibitors. We also hypothesized that medical conditions, including blood pressure, were well-controlled, at least during chemotherapy, because the enrolled patients were undergoing adjuvant chemotherapy. Third, clinical outcomes according to the long-term use of RAS inhibitors were not estimated in this study. We aimed to evaluate the efficacy of RAS inhibitors against drug toxicity during DOX administration; therefore, use of RAS inhibitors was investigated for 3 months from the index date, which is the approximate mean duration of DOX administration that includes 4 cycles of DOX dosing. Fourth, the sample size of RAS inhibitor group, which was sub-classified from the HT group, was relatively small. According to the Korean hypertension fact sheet 2022 published by the Korean Society of Hypertension [44], patients who received RAS inhibitors as monotherapy comprised approximately 20% of the total Korean population with hypertension. This trend in the treatment of hypertension is consistent with that observed in our study.

## Conclusions

Hypertension is a risk factor for HF in patients with breast cancer undergoing DOX-containing chemotherapy. However, there is no clear evidence that hypertension increases overall mortality, and therefore it does not necessarily worsen clinical outcomes in patients with cancer therapy. Conversely, the use of RAS inhibitors for treating hypertension may contribute to decreased mortality and improved clinical outcomes, although the protective effects of RAS inhibitors against DOX-related HF remain unclear. Therefore, careful management of hypertension, including considering the use of RAS inhibitors, is required for patients with hypertension and breast cancer undergoing DOX therapy.

## Supporting information

**S1 Table. Definitions of comorbidities, medications, and clinical outcomes.**
(DOCX)

**S2 Table. Covariates for propensity score matched cohorts.**
(DOCX)

**S3 Table. Mean follow-up and differential anticancer therapies between the two groups after propensity score matching.**
(DOCX)

**S4 Table. Sensitivity test: The 1-year primary outcome for each propensity score-matched cohort.**
(DOCX)

## Acknowledgments

The authors would like to thank Hun Hee Lee, Master of Arts in Statistics, Healthcare Big-Date Center, Research Institute of Clinical Medicine, Kyung Hee University Hospital at Gang Dong for assistance with the statistical analysis.

## Author Contributions

**Conceptualization:** Hui-Jeong Hwang.

**Data curation:** Hui-Jeong Hwang.

**Formal analysis:** Hui-Jeong Hwang, Taek-Gu Lee.

**Funding acquisition:** Hui-Jeong Hwang.

**Investigation:** Hui-Jeong Hwang.

**Methodology:** Hui-Jeong Hwang, Taek-Gu Lee.

**Supervision:** Taek-Gu Lee.

**Validation:** Hui-Jeong Hwang.

**Writing – original draft:** Hui-Jeong Hwang.

**Writing – review & editing:** Taek-Gu Lee.

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
