## [Decision Letter · Decision Letter 0]

12 Oct 2023

PONE-D-23-27903Impact on Clinical Outcomes of Renin-Angiotensin System Inhibitors Against Doxorubicin-Related Toxicity in Patients with Breast Cancer and Hypertension: A Nationwide Cohort Study in South KoreaPLOS ONE

Dear Dr. Lee,

Thank you for submitting your manuscript to PLOS ONE. After careful consideration, we feel that it has merit but does not fully meet PLOS ONE’s publication criteria as it currently stands. Therefore, we invite you to submit a revised version of the manuscript that addresses the points raised during the review process.

We look forward to receiving your revised manuscript.

Kind regards,

Amirmohammad Khalaji

Academic Editor

PLOS ONE

-  https://doi.org/10.4048/jbc.2023.26.e34

In your revision ensure you cite all your sources (including your own works), and quote or rephrase any duplicated text outside the methods section. Further consideration is dependent on these concerns being addressed.

Reviewers' comments:

Reviewer's Responses to Questions

**Comments to the Author**

1. Is the manuscript technically sound, and do the data support the conclusions?

Reviewer #1: Yes

Reviewer #2: Yes

2. Has the statistical analysis been performed appropriately and rigorously? 

Reviewer #1: Yes

Reviewer #2: Yes

3. Have the authors made all data underlying the findings in their manuscript fully available?

Reviewer #1: Yes

Reviewer #2: No

4. Is the manuscript presented in an intelligible fashion and written in standard English?

Reviewer #1: Yes

Reviewer #2: Yes

5. Review Comments to the Author

Reviewer #1: The current study has interesting and practical topic, discussing a fundamental issue and it also adds valuable data to current available knowledge. I found the paper to be overall well written.

The abstract clearly states the aim of the study and also methods and results.

The introduction contains clear information about the topic and, cites the most recent, relevant studies explaining the controversy in existing articles.

Inclusion and exclusion criteria, the sampling method and results are wisely done and I can’t find any issues in them. The figures and tables demonstrate essential data in a basic and comprehensible manner.

The discussion has a coherent schema that perfectly mentions conclusions aligned with methods and results in an arranged structure. The interpretation reasonably reflects the results.

Limitations section includes statements that were missed in the exclusion criteria.

Reviewer #2: Lee et al. have performed a study on effect of RAS inhibitors on doxorubicin-related toxicity in patients with breast cancer and hypertension. By investigating 54,344 patients, they showed that RAS inhibition can lead to better prognosis in those with breast cancer and hypertension undergoing doxorubicin therapy. The findings are interesting, however, there are some minor comments to apply:

- In background section of the abstract, it is highly suggested to provide a sentence for use of doxorubicin in breast cancer.

- Do the authors have any data regarding the stage of cancer in these patients? Was any limitation applied regarding the staging of breast cancer?

- The definition of standard DOX therapy should be mentioned and cited in methods section. What was the dosage? Was it constant in all patients?

- Line 117: There is no need to mention the time analysis was performed.

- What is the take home message of this study for clinicians? It could be added as a paragraph to the discussion section.

6. PLOS authors have the option to publish the peer review history of their article (what does this mean?). If published, this will include your full peer review and any attached files.

Reviewer #1: No

Reviewer #2: No

---

## [Author Response · Author response to Decision Letter 0]

1 Nov 2023

Dear Academic Editor Amirmohammad Khalaji

We are very pleased to send a transmittal letter of our revision.

We answered and revised our manuscript about reviewer’s and editorial comments. 

We really thank you for considering it for publication and hope this manuscript can be accepted.

[Journal Requirements]

Answer: We revised the manuscript according the style requirements of PLOS ONE.

https://doi.org/10.4048/jbc.2023.26.e34

In your revision ensure you cite all your sources (including your own works), and quote or rephrase any duplicated text outside the methods section. Further consideration is dependent on these concerns being addressed.

Answer: In our previous research, as mentioned above, we compared clinical outcomes among patients with hypertension and breast cancer undergoing DOX therapy who were treated with different antihypertensive drugs. Conversely, the objective of this study is to compare adverse outcomes related to DOX treatment between patients with and without hypertension. Therefore, we initially believed that there would be little relevance between the two studies. However, as per your comments, since they share a similar methodology, we have included the reference and revised the article in the introduction. 

Revised:

(line 67, page 4) In our recent study, RAS inhibitors had better cardiovascular outcomes than beta blockers or thiazide and thiazide-like diuretics in patients with hypertension and breast cancer undergoing DOX therapy, but similar outcomes to calcium channel blockers [18]. 

(line 86, page 5) As described in our previous study [18], female patients aged 17–70 years who were first diagnosed with breast cancer and started DOX therapy between January 1, 2008 and December 31, 2015 were recruited (Fig 1). The index date was defined as the date on which DOX treatment was initiated.

Answer: We reviewed and revised the references list.

[Reviewers' comments]

Reviewer #1:

 The current study has interesting and practical topic, discussing a fundamental issue and it also adds valuable data to current available knowledge. I found the paper to be overall well written. The abstract clearly states the aim of the study and also methods and results. The introduction contains clear information about the topic and, cites the most recent, relevant studies explaining the controversy in existing articles. Inclusion and exclusion criteria, the sampling method and results are wisely done and I can’t find any issues in them. The figures and tables demonstrate essential data in a basic and comprehensible manner. The discussion has a coherent schema that perfectly mentions conclusions aligned with methods and results in an arranged structure. The interpretation reasonably reflects the results.

Limitations section includes statements that were missed in the exclusion criteria.

Answer: Thank you for your good estimation.

Reviewer #2

1. In background section of the abstract, it is highly suggested to provide a sentence for use of doxorubicin in breast cancer.

Answer: Thank you for your comments. We revised the abstract to include the sentence associated with usual use of doxorubicin in breast cancer.

Revised: (line 21, page 2) Background: Although doxorubicin (DOX) is a commonly used potent chemotherapeutic agent in patients with breast cancer, its cardiotoxic effect is a concern, particularly in patients with hypertension. Antihypertensive renin-angiotensin system (RAS) inhibitors may potentially play a role in preventing overt heart failure (HF) due to DOX toxicity.

2. Do the authors have any data regarding the stage of cancer in these patients? Was any limitation applied regarding the staging of breast cancer?

Answer: As previously mentioned in the limitations section, we were unable to obtain data related to cancer stage. Instead, we excluded patients with metastasis to minimize the impact of cancer staging on mortality. According to statistical publications from the Korean Breast Cancer Society (https://www.kbcs.or.kr), the 5-year survival rate of Korean patients with breast cancer between 2001 and 2012 dropped rapidly to below 35% in stage IV, compared with those in stages I-III (higher than 90% in stages I-II, 75% in stage III). We have additionally included this information in the limitations section for clarity. 

Revised: (line 263, page 15) Second, the HIRA Service database of South Korea does not provide clinical data, including cancer stage and blood pressure. According to statistical reports from the Korean Breast Cancer Society (https://www.kbcs.or.kr), the 5-year survival rates for Korean patients with breast cancer between 2001 and 2012 dropped rapidly to below 35% in stage IV, compared with those in stages I-III (above 90% in stages I-II, 75% in stage III). Therefore, we excluded patients with metastasis to minimize mortality unrelated to interventions, including chemotherapy and RAS inhibitors.

3. The definition of standard DOX therapy should be mentioned and cited in methods section. What was the dosage? Was it constant in all patients?

Answer: As far as I know, it is challenging to clearly define standard DOX therapy in terms of dosing or frequency. In the adjuvant and neoadjuvant settings, DOX-containing chemotherapy regimens usually recommend administering a dosage of 60 mg/m2 per cycle for a total of 4 cycles of DOX. Moreover, several guidelines, including those from the ESC (European Society of Cardiology) and ASCO (American Society of Clinical Oncology), consider ≥250 mg/m2 of doxorubicin as a higher risk. Therefore, we defined high-dose DOX as > 4 cycles of DOX or a cumulative dose of > 240 mg/m2. According to this definition, the total amount of DOX used in each patient was calculated and then divided into 'yes' or 'no' as a covariate for high-dose DOX, which was adjusted as a confounding factor for clinical outcomes. We additionally dictated it in the method section.

Revised: (line 114, page 6) The total amount of DOX used in each patient was calculated and high-dose DOX was defined as > 4 cycles of DOX or a cumulative dose of > 240 mg/m2 [1].

4. Line 117: There is no need to mention the time analysis was performed.

Answer: Thank you for your comment. We removed the sentence.

5. What is the take home message of this study for clinicians? It could be added as a paragraph to the discussion section.

Answer: As per your comments, we summarized the results of this study in the first paragraph of the discussion and emphasized the clinical implications of the findings in the conclusion section, which inform the take-home message of this study. 

Revised: (line 201, page 12) Hypertension is a concurrent risk factor for HF [19,20] and DOX-related cardiotoxicity [1,3,4]. However, there is no evidence that hypertension increases mortality in patients with cancer [21]. In this study, patients with hypertension had a higher risk of HF than those without hypertension, but the overall mortality was similar in both the groups. Consequently, the primary outcome was not significantly different. In contrast, patients taking RAS inhibitors to treat hypertension had a risk of HF similar to those without hypertension, but had a lower mortality rate and consequently a lower primary outcome.

(line 284, page 15) Hypertension is a risk factor for HF in patients with breast cancer undergoing DOX-containing chemotherapy. However, there is no clear evidence that hypertension increases overall mortality, and therefore it does not necessarily worsen clinical outcomes in patients with cancer therapy. Conversely, the use of RAS inhibitors for treating hypertension may contribute to decreased mortality and improved clinical outcomes, although the protective effects of RAS inhibitors against DOX-related HF remain unclear. Therefore, careful management of hypertension, including considering the use of RAS inhibitors, is required for patients with hypertension and breast cancer undergoing DOX therapy.

---

## [Editor Report · Decision Letter 1]

7 Nov 2023

Impact on clinical outcomes of renin-angiotensin system inhibitors against doxorubicin-related toxicity in patients with breast cancer and hypertension: a nationwide cohort study in South Korea

PONE-D-23-27903R1

Dear Dr. Lee,

We’re pleased to inform you that your manuscript has been judged scientifically suitable for publication and will be formally accepted for publication once it meets all outstanding technical requirements.

Kind regards,

Amirmohammad Khalaji

Academic Editor

PLOS ONE
---

## [Editor Report · Acceptance letter]

10 Nov 2023

PONE-D-23-27903R1 

Impact on clinical outcomes of renin-angiotensin system inhibitors against doxorubicin-related toxicity in patients with breast cancer and hypertension: a nationwide cohort study in South Korea 

Dear Dr. Lee:

I'm pleased to inform you that your manuscript has been deemed suitable for publication in PLOS ONE. Congratulations! Your manuscript is now with our production department. 

Kind regards, 

on behalf of

Dr. Amirmohammad Khalaji 

Academic Editor

PLOS ONE